statistical physics/statistics

temporal network, power-law distribution, Poisson process, model selection

**Author for correspondence:**
Naoki Masuda
e-mail: naokimas@buffalo.edu

# Long-tailed distributions of inter-event times as mixtures of exponential distributions

Makoto Okada[1], Kenji Yamanishi[1] and Naoki Masuda[2,3,4]

[1]Graduate School of Information Science and Technology, The University of Tokyo, 7-3-1 Hongo Bunkyo, Tokyo 113-8656, Japan
[2]Department of Engineering Mathematics, University of Bristol, Woodland Road, Clifton, Bristol BS8 1UB, UK
[3]Department of Mathematics, University at Buffalo, State University of New York, Buffalo, NY 14260-2900, USA
[4]Computational and Data-Enabled Science and Engineering Program, University at Buffalo, State University of New York, Buffalo, NY 14260-5030, USA

NM, 0000-0003-1567-801X

Inter-event times of various human behaviour are apparently non-Poissonian and obey long-tailed distributions as opposed to exponential distributions, which correspond to Poisson processes. It has been suggested that human individuals may switch between different states, in each of which they are regarded to generate events obeying a Poisson process. If this is the case, inter-event times should approximately obey a mixture of exponential distributions with different parameter values. In the present study, we introduce the minimum description length principle to compare mixtures of exponential distributions with different numbers of components (i.e. constituent exponential distributions). Because these distributions violate the identifiability property, one is mathematically not allowed to apply the Akaike or Bayes information criteria to their maximum-likelihood estimator to carry out model selection. We overcome this theoretical barrier by applying a minimum description principle to joint likelihoods of the data and latent variables. We show that mixtures of exponential distributions with a few components are selected, as opposed to more complex mixtures in various datasets, and that the fitting accuracy is comparable to that of state-of-the-art algorithms to fit power-law distributions to data. Our results lend support to Poissonian explanations of apparently non-Poissonian human behaviour.

## 1. Introduction

Many social and economic processes are a consequence of human behaviour. Technological advances are increasingly enabling us to

record various human behaviours in quantitative manners. Such quantitative understanding often challenges traditional assumptions underlying mathematical modelling of human behaviour, a major instance of which is a lack of Poissonian properties in a range of human behavioural data. In other words, when a sequence of time-stamped events, such as email correspondence, is observed from a human individual, the event sequence often deviates from Poisson processes, in which inter-event times independently obey an exponential distribution. Rather, empirical inter-event times from human behaviour and other data often obey long-tailed distributions [1,2]. Such non-Poissonian processes on nodes and edges are building blocks of temporal networks on which dynamical processes such as epidemic processes often behave differently from the same processes on static networks [3,4].

Two classes of models that generate long-tailed distributions of inter-event times are priority queue models and modulated Markov processes [3,4]. In priority queue models, one assumes that a human individual is a queue that receives tasks across time and prioritizes some particular tasks for execution [1]. By contrast, in modulated Markov processes, one assumes that a human individual generates events according to a Poisson process (i.e. one determines whether to have an event right now at a constant rate without memory) but modulates the event rate. The event rate may be assumed to take one of a few values [5–10] or continuously many values [11,12]. When we assume a few states, where each state corresponds to a Poisson process, an underlying assumption about human behaviour is that an individual transits among a few distinguishable states, such as an active state and an inactive state. If this is the case, we should be able to fit a mixture of exponential distributions rather than power-law distributions to empirical distributions of inter-event times with a reasonable accuracy. This is because a subset of inter-event times, i.e. those produced in each state, is expected to obey an exponential distribution.

In fact, a mixture of a small number of exponential distributions and a power-law distribution with an exponential cut-off can look similar. In the present study, we fit mixtures of exponential distributions to several datasets of inter-event times. The idea of approximating empirical distributions resembling power-law distributions by mixtures of exponential distributions is not a new idea [13]. We perform model selection to determine the number of component exponential distributions fitted to data and also compare the mixture of exponential distributions with two types of conventionally used power-law distributions. A technical challenge is that the maximum-likelihood estimator of the mixture of exponential distributions does not satisfy the asymptotic normality such that it is not allowed to use the Akaike information criterion (AIC) or Bayes information criterion (BIC) (e.g. [14, p. 203]). Therefore, based on the minimum description length (MDL) criterion [15,16], we perform the procedure so-called latent variable completion [17–20] to derive variants of MDL justified for mixtures of exponential distributions. We find that mixtures of up to three exponential distributions are the best performer in many cases. Python codes for estimating and selecting the mixture of exponential distributions are available at Github (https://github.com/naokimas/exp_mixture_model).

## 2. Methods

### 2.1. Mixture of exponential distributions and its maximum-likelihood estimation

We fit mixtures of exponential distributions, which we refer to as the exponential mixture models, abbreviated as EMMs, to distributions of inter-event times, denoted by $\tau \in [0, \infty)$, and compare the fit with the case of power-law distributions. We denote by $k$ the number of the exponential distributions to be mixed, which we refer to as the number of component distributions. The mixing weight, i.e. the probability that the $j$th exponential distribution is used, is denoted by $\pi_j$ ($1 \leq j \leq k$). The probability density function for an EMM is given by

$$p(\tau; \boldsymbol{\pi}, \boldsymbol{\mu}) = \sum_{j=1}^{k} \frac{\pi_j}{\mu_j} \exp\left(-\frac{\tau}{\mu_j}\right), \tag{2.1}$$

where $\boldsymbol{\pi} \equiv \{\pi_1, \ldots, \pi_k\}$, $\boldsymbol{\mu} \equiv \{\mu_1, \ldots, \mu_k\}$ and $\mu_j$ is the mean of the $j$th exponential distribution.

Given a series of inter-event times, $\boldsymbol{\tau} \equiv \{\tau_1, \ldots, \tau_n\}$, and the number of components, $k$, we estimate the values of the model parameters, $\theta = (\boldsymbol{\pi}, \boldsymbol{\mu})$, as follows.

First, we run the expectation-maximization (EM) algorithm [21]. As an initial condition, we set $\pi_j^{(0)} = 1/k$ ($1 \leq j \leq k$), where the superscript indicates the iteration number in the EM procedure. We also draw $\mu_j^{(0)}$ ($1 \leq j \leq k$) such that $\log_{10} \mu_j^{(0)}$ independently obeys the uniform density on $[\log_{10} \tau_{\min},$

$\log_{10}\tau_{\max}]$, where $\tau_{\min}$ and $\tau_{\max}$ are the minimum and maximum of the inter-event time in the given dataset, respectively. In this manner, we intend to generate a sufficiently broad initial distribution of $\mu_j^{(0)}$ while a majority of the $\mu_j^{(0)}$ values is relatively small. Intuitively, a large value of $\mu_j^{(0)}$, which sometimes occurs, reflects the long tail of the distribution of inter-event times. We iterate the following EM steps $t_{\max}$ times by alternating the expectation (E) and maximization (M) steps. The E step is given by

$$\gamma_{ij}^{(t)} = \frac{\pi_j^{(t)} p(\tau_i; \mu_j^{(t)})}{\sum_{j'=1}^{k} \pi_{j'}^{(t)} p(\tau_i; \mu_{j'}^{(t)})}, \tag{2.2}$$

where $t = 0, 1, 2, \ldots, t_{\max} - 1$. One can interpret $\gamma_{ij}^{(t)}$ as the probability that $\tau_i$ is generated from the $j$th exponential distribution. The M step is given by

$$\pi_j^{(t+1)} = \frac{1}{n} \sum_{i=1}^{n} \gamma_{ij}^{(t)} \tag{2.3a}$$

and

$$\mu_j^{(t+1)} = \frac{\sum_{i=1}^{n} \gamma_{ij}^{(t)} \tau_i}{\sum_{i=1}^{n} \gamma_{ij}^{(t)}}. \tag{2.3b}$$

After the iteration of the E and M steps $t_{\max}$ times, one obtains $\pi_1^{(t_{\max})}, \ldots, \pi_k^{(t_{\max})}$ and $\mu_1^{(t_{\max})}, \ldots, \mu_k^{(t_{\max})}$, which is an approximate maximum-likelihood estimator. This estimator corresponds to the case in which one does not estimate the latent variables that explicitly indicate which exponential distribution out of the $k$ exponential distributions has generated each $\tau_i$.

When the values of the latent variables are required, we compute them as follows. We denote the latent variable corresponding to each $\tau_i (1 \leq i \leq n)$ by $z_i (1 \leq z_i \leq k)$. In other words, $\tau_i$ is estimated to be generated from the $z_i$th exponential distribution. We first estimate the latent variables by

$$\hat{z}_i = \arg\max_{j=1,\ldots,k} \gamma_{ij}^{(t_{\max})}$$
$$= \arg\max_{j=1,\ldots,k} [\pi_j^{(t_{\max})} p(\tau_i; \mu_j^{(t_{\max})})]. \tag{2.4}$$

We then estimate the other parameters by

$$\hat{\pi}_j(\hat{z}) = \frac{n_j}{n}, \tag{2.5}$$

where $\hat{z} = \{\hat{z}_1, \ldots, \hat{z}_n\}$, $n_j$ is the number of inter-event times whose estimated latent variable $\hat{z}_i$ is equal to $j$, and

$$\hat{\mu}_j(\tau, \hat{z}) = \frac{1}{n_j} \sum_{i=1:\hat{z}_i=j}^{n} \tau_i. \tag{2.6}$$

The $\hat{\pi}_j$ and $\hat{\mu}_j$ values given by equations (2.5) and (2.6) are a maximum-likelihood estimator when the joint distribution of the inter-event times (i.e. $\tau_1, \ldots, \tau_n$) and the latent variables (i.e. $\hat{z}_1, \ldots, \hat{z}_n$) is estimated. They are different from the values estimated by the EM algorithm (i.e. $\pi_1^{(t_{\max})}, \ldots, \pi_k^{(t_{\max})}$ and $\mu_1^{(t_{\max})}, \ldots, \mu_k^{(t_{\max})}$).

To cope with the problem of local maxima, we estimate the values of $\hat{\pi}_j$, $\hat{\mu}_j$, and $\hat{z}_i$ $(1 \leq j \leq k, 1 \leq i \leq n)$ 10 times starting from different initial conditions. Then, among the 10 maximum-likelihood estimators, we employ the one that has yielded the largest likelihood of the joint distribution of $\tau$ and $\hat{z}$.

## 2.2. Model selection criteria for mixtures of exponential distributions

We consider the following six model selection criteria.

### 2.2.1. AIC and BIC

For an EMM with $k$ components without an explicit consideration of the latent variables, the AIC [22] and the BIC [23] are given by

$$\text{AIC}(\tau) = -\log p(\tau; \hat{\theta}_{\text{EM}}(\tau)) + 2k - 1 \tag{2.7}$$

and

$$\mathrm{BIC}(\boldsymbol{\tau}) = -\log p(\boldsymbol{\tau}, \hat{\theta}_{\mathrm{EM}}(\boldsymbol{\tau})) + \frac{2k-1}{2}\log n, \tag{2.8}$$

respectively. In equations (2.7) and (2.8), $\hat{\theta}_{\mathrm{EM}}(\boldsymbol{\tau})$ is the EM estimator of the parameters of the EMM, i.e. $\pi_1^{(t_{\max})}, \ldots, \pi_k^{(t_{\max})}, \mu_1^{(t_{\max})}, \ldots, \mu_k^{(t_{\max})}$.

### 2.2.2. AIC and BIC with latent variable completion

An EMM is non-identifiable, which by definition dictates that a distribution does not correspond to a parameter set in a one-to-one manner [14]. For example, if $k=2$ and $\mu_1 = \mu_2$, the distribution of $\tau$ is the same exponential distribution regardless of the value of $\pi_1(=1-\pi_2)$. When a distribution is non-identifiable, the maximum-likelihood estimator is not asymptotically normal, and the conventional AIC and BIC, which are derived on the basis of asymptotic normality, lose justification [14].

A method for overcoming this theoretical barrier is to apply model selection criteria to a complete variable model, in which one completes the values of the latent variables, rather than to use a marginalized model [17–20]. We call this method the latent variable completion [20] throughout this paper. Under latent variable completion, one explicitly incorporates the likelihood of the latent variables, $z_i(1 \leq i \leq n)$, without marginalizing them, into a model selection criterion. Then, the joint distribution of $\tau_i$ and $z_i(1 \leq i \leq n)$ is an identifiable distribution such that the use of the AIC and BIC is justified. Also in practice, latent variable completion is better at describing some empirical data (for example, at estimating an appropriate number of components, $k$) than model selection criteria that do not use latent variable completion [17,18].

We refer to the corresponding AIC or BIC as the AIC or BIC with latent variable completion [17,19] and denote them by $\mathrm{AIC_{LVC}}$ and $\mathrm{BIC_{LVC}}$, respectively. We obtain

$$\mathrm{AIC_{LVC}}(\boldsymbol{\tau}, \hat{z}) = -\log p(\boldsymbol{\tau}, \hat{z}; \hat{\theta}(\boldsymbol{\tau}, \hat{z})) + 2k^* - 1 \tag{2.9}$$

and

$$\mathrm{BIC_{LVC}}(\boldsymbol{\tau}, \hat{z}) = -\log p(\boldsymbol{\tau}, \hat{z}; \hat{\theta}(\boldsymbol{\tau}, \hat{z})) + \frac{k^* - 1}{2}\log n + \frac{1}{2}\sum_{j=1}^{k^*}\log n_j. \tag{2.10}$$

In equations (2.9) and (2.10), $\hat{\theta}(\boldsymbol{\tau}, \hat{z}) = \{\hat{\pi}_1(\hat{z}), \ldots, \hat{\pi}_{k^*}(\hat{z}), \hat{\mu}_1(\boldsymbol{\tau}, \hat{z}), \ldots, \hat{\mu}_{k^*}(\boldsymbol{\tau}, \hat{z})\}$, derived in equations (2.5) and (2.6), is the maximum-likelihood estimator for the joint distribution of inter-event times $\boldsymbol{\tau}$ and latent variables $\hat{z}$. The likelihood in equations (2.9) and (2.10) is given by

$$p(\boldsymbol{\tau}, \hat{z}; \hat{\theta}(\boldsymbol{\tau}, \hat{z})) = \prod_{j=1}^{k^*}(\hat{\pi}_j)^{n_j}p(\{\tau_i\}_{i=1, \hat{z}_i=j}^{n}; \hat{\mu}_j) = \prod_{j=1}^{k^*}\left(\frac{\hat{\pi}_j}{e\hat{\mu}_j}\right)^{n_j}. \tag{2.11}$$

In equations (2.9), (2.10) and (2.11), $k^*$ is the number of components that are used at least once under the estimated latent variable values. The latent variable specifies which of the $k$ exponential distributions has produced $\tau_i$ for each $i$. Therefore, as a result of estimating the latent variables, a $j$th exponential distribution may not have any inter-event times $\tau_i$ belonging to it (i.e. $n_j = 0$). We exclude such exponential distributions, i.e. those with $n_j = 0$, and only consider the remaining $k^*$ exponential distributions. To calculate $\mathrm{AIC_{LVC}}$ and $\mathrm{BIC_{LVC}}$, we use $k^*$ instead of $k$ because $k^*$ is the actual number of components given the estimated latent variable values. For the same reason, we will use $k^*$ instead of $k$ in the two model selection criteria introduced in the following sections, which also explicitly use the estimated latent variables and latent variable completion.

### 2.2.3. Normalized maximum-likelihood codelength

Here, we introduce model selection criteria on the basis of the MDL principle [16]. This principle asserts that the best model should be the one that attains the shortest codelength required for encoding the data as well as the model itself. The codelength is the number of bits required for encoding the data into a binary sequence under the information-theoretic requirement that each codeword can be uniquely decodable even without commas. In fact, MDL approaches to mixture modelling have a long history, as represented by the Snob program [24,25].

The normalized maximum-likelihood (NML) codelength is a type of MDL [16,26]. The NML codelength minimizes the worst-case (i.e. in terms of data $\boldsymbol{x}$) regret value, which is equal to the actual

codelength minus the ideal codelength, i.e. $\min_{\theta \in \Theta}(-\log p(x; \theta))$ [27]. The other two model selection criteria, which we will explain in the next two sections, are based on the NML. Therefore, we explain the NML codelength in this section. Although we assume that inter-event times $\tau_i$ are continuous-valued, we first explain the NML codelength for a sequence of discrete-valued variables $x = \{x_1, \ldots, x_n\}$ for expository purposes.

To shorten the codelength for the given data $x$, one should in principle minimize the Shannon information given by $-\log p(x; \theta)$, where we assume throughout the present paper that log is in base $e$ unless we specify the base. The minimizer is obviously given by the maximum-likelihood estimator, $\hat{\theta}(x)$. However, the maximum-likelihood estimator generally yields

$$\sum_x p(x; \hat{\theta}(x)) > 1, \tag{2.12}$$

because $\hat{\theta}$ depends on $x$. In equation (2.12), the summation is taken over all possible values of $x$. Therefore, we use the normalized probability distribution given by

$$p_{\text{NML}}(x) = \frac{p(x; \hat{\theta}(x))}{\sum_{x'} p(x'; \hat{\theta}(x'))} \tag{2.13}$$

to encode the data [28]. In other words, we set

$$\begin{aligned} L_{\text{NML}}(x) &= -\log p_{\text{NML}}(x) \\ &= -\log p(x; \hat{\theta}(x)) + \log C(n), \end{aligned} \tag{2.14}$$

where

$$C(n) \equiv \sum_{x'} p(x'; \hat{\theta}(x')) \tag{2.15}$$

is called the parametric complexity. In equation (2.14), the first term on the right-hand side is small when the model fits the data well, and the second term represents the complexity of the model. In general, $C(n)$ tends to increase as $k$ increases because an increase in $k$ leads to the expansion of the parameter space in which one searches $\hat{\theta}$, which leads to an increase in $C(n)$ [29].

One can similarly derive the NML codelength for a sequence of continuous-valued variables by replacing the summation by the integral. For example, the equivalent of equation (2.15) in the case of continuous-valued variables is given by

$$C(n) = \int p(x'; \hat{\theta}(x')) \, dx'_1 \cdots dx'_n. \tag{2.16}$$

In the remainder of this section, we explain the NML codelength for an exponential distribution [19,30], not for an EMM, for two reasons. First, analytically calculating the NML codelength for an EMM seems formidable. Second, we will use the NML codelength for single exponential distributions in deriving the two types of the NML-based codelengths for EMMs in the following sections.

Consider an exponential distribution given by

$$p(\tau; \mu) = \frac{1}{\mu} \exp\left(-\frac{\tau}{\mu}\right). \tag{2.17}$$

For this distribution, we obtain

$$L_{\text{NML}}(\boldsymbol{\tau}) = -\log p(\boldsymbol{\tau}; \hat{\mu}(\boldsymbol{\tau})) + \log C_{\text{exp}}(n) \tag{2.18a}$$

and

$$C_{\text{exp}}(n) = \int_{\mathcal{T}(\mu_{\min}, \mu_{\max})} p(\boldsymbol{\tau}'; \hat{\mu}(\boldsymbol{\tau}')) \, d\tau'_1 \cdots d\tau'_n, \tag{2.18b}$$

where $\mathcal{T}(\mu_{\min}, \mu_{\max})$ is a region of $\boldsymbol{\tau}' \equiv \{\tau'_1, \ldots, \tau'_n\} \in \mathbf{R}^n$ such that the maximum-likelihood estimator given by

$$\hat{\mu} = \frac{1}{n} \sum_{i=1}^{n} \tau'_i \tag{2.19}$$

is contained in $[\mu_{\min}, \mu_{\max}]$. We have to confine the domain of integration to $\mathcal{T}$ in this manner because the

integral in equation (2.18b) diverges if $\mathcal{T}$ is replaced by $\mathbf{R}^n$. In practice, if we take $\mu_{\min}\,(>0)$ small enough and $\mu_{\max}$ large enough, $\hat{\mu} \in [\mu_{\min}, \mu_{\max}]$ would be satisfied for empirical sequences of inter-event times $\tau'$. The standard method to evaluate the integral in equation (2.18b) is to transform the integral to that in the parameter space [19,31]. This method yields an integral of the form similar to $\int_{\mu_{\min}}^{\mu_{\max}} (1/\hat{\mu}) \mathrm{d}\hat{\mu}$, which is manageable [19]. Region $\mathcal{T}$ is a useful heuristic for avoiding such a divergence in NML-related calculations [31].

To incorporate the codelength necessary for encoding the value of $\mu_{\min}$ and $\mu_{\max}$ into the NML codelength, we rewrite $\mu_{\min} = \exp(m_{\min})$ and $\mu_{\max} = \exp(m_{\max})$, where $m_{\min}$ and $m_{\max}$ are integers. Then, we encode $m_{\min}$ and $m_{\max}$ instead of $\mu_{\min}$ and $\mu_{\max}$.

The first term on the right-hand side of equation (2.18a) is given by

$$
\begin{aligned}
\log p(\boldsymbol{\tau}; \hat{\mu}(\boldsymbol{\tau})) &= \log \prod_{i=1}^{n} \frac{1}{\hat{\mu}} \exp\left(-\frac{\tau_i}{\hat{\mu}}\right) \\
&= -n \log \hat{\mu} - \frac{1}{\hat{\mu}} \sum_{i=1}^{n} \tau_i \\
&= -n \log \hat{\mu} - n,
\end{aligned}
\tag{2.20}
$$

where $\hat{\mu}$ is given by equation (2.19). To obtain the second term on the right-hand side of equation (2.18a), we substitute equation (2.17) into equation (2.18b), which leads to

$$
\begin{aligned}
\log C_{\exp}(n) &= n \log n - n - \log \Gamma(n) + \log \log \frac{\mu_{\max}}{\mu_{\min}} \\
&= n \log n - n - \log \Gamma(n) + \log(m_{\max} - m_{\min}),
\end{aligned}
\tag{2.21}
$$

where $\Gamma(n)$ is the gamma function.

By substituting equations (2.20) and (2.21) into equation (2.18a), one obtains

$$
L_{\mathrm{NML}}(\boldsymbol{\tau}) = n \log \hat{\mu} + n \log n - \log \Gamma(n) + \log(m_{\max} - m_{\min}).
\tag{2.22}
$$

To carry out model selection, we add the codelength for $m_{\min}$ and $m_{\max}$, denoted by $\ell(m_{\min})$ and $\ell(m_{\max})$, to $L_{\mathrm{NML}}(\boldsymbol{\tau})$ to obtain

$$
\tilde{L}_{\mathrm{NML}}(\boldsymbol{\tau}) = L_{\mathrm{NML}}(\boldsymbol{\tau}) + \ell(m_{\min}) + \ell(m_{\max}).
\tag{2.23}
$$

The derivation of $\ell(m_{\min})$ and $\ell(m_{\max})$ is given in the electronic supplementary material.

### 2.2.4. Normalized maximum-likelihood codelength with latent variable completion

As a first NML type of the model selection criterion for EMMs, we consider a latent variable completion of the NML codelength. We refer to the resulting criterion by $\mathrm{NML}_{\mathrm{LVC}}$ and the codelength by $L_{\mathrm{LVC}}$. Although an analytical expression for the NML codelength for an EMM without latent variable completion is unavailable, we can analytically calculate the NML codelength for the joint distribution of inter-event times and the latent variables.

Similar to the derivation in the case of a mixture of Gaussian distributions [19], we derive the codelength for EMMs as follows:

$$
L_{\mathrm{LVC}}(\boldsymbol{\tau}, \hat{\boldsymbol{z}}) = -\log p(\boldsymbol{\tau}, \hat{\boldsymbol{z}}; \hat{\theta}(\boldsymbol{\tau}, \hat{\boldsymbol{z}})) + \log C_{\mathrm{EMM}}(n, k^*),
\tag{2.24}
$$

where

$$
C_{\mathrm{EMM}}(n, k^*) = \sum_{\hat{z}_1'=1}^{k^*} \cdots \sum_{\hat{z}_n'=1}^{k^*} \int_{\mathcal{T}'(\mu_{\min}', \mu_{\max}')} p(\boldsymbol{\tau}', \hat{\boldsymbol{z}}'; \hat{\theta}(\boldsymbol{\tau}', \hat{\boldsymbol{z}}')) \, \mathrm{d}\tau_1' \cdots \mathrm{d}\tau_n'.
\tag{2.25}
$$

We remind that $p(\boldsymbol{\tau}', \hat{\boldsymbol{z}}'; \hat{\theta}(\boldsymbol{\tau}', \hat{\boldsymbol{z}}'))$ is given by equation (2.11). Region $\mathcal{T}'(\mu_{\min}', \mu_{\max}')$ of $\boldsymbol{\tau}' \in \mathbf{R}^n$ is defined such that each of $\hat{\mu}_1, \ldots, \hat{\mu}_{k^*}$ in the maximum-likelihood estimator $\hat{\theta}(\boldsymbol{\tau}', \hat{\boldsymbol{z}}')$ when the latent variables are explicitly estimated (i.e. equation (2.6)) is contained in $[\mu_{\min}', \mu_{\max}']$. Region $\mathcal{T}'(\mu_{\min}', \mu_{\max}')$ coincides with region $\mathcal{T}(\mu_{\min}', \mu_{\max}')$ when $k^* = 1$. Similarly to the case of equation (2.18b), we confine the domain of integration to $\mathcal{T}'$ to avoid the divergence of the integral in equation (2.25). We set

$$
\mu_{\min}' = \exp(m_{\min}'),
\tag{2.26}
$$

where

$$m'_{\min} = \left\lfloor \log\left(\min_{j=1,\ldots,k^*} \hat{\mu}_j\right)\right\rfloor, \tag{2.27}$$

and

$$\mu'_{\max} = \exp(m'_{\max}), \tag{2.28}$$

where

$$m'_{\max} = \left\lceil \log\left(\max_{j=1,\ldots,k^*} \hat{\mu}_j\right)\right\rceil, \tag{2.29}$$

for two reasons. First, $\log C_{\mathrm{EMM}}(n, k^*)$ is small when $\mu'_{\min}$ is large or $\mu'_{\max}$ is small. Second, encoding of $\mu'_{\min}$ and $\mu'_{\max}$ is facilitated by the introduction of integer variables, as we did for the NML codelength (§2.2.3).

By substituting

$$-\log p(\boldsymbol{\tau}, \hat{z}; \hat{\theta}(\boldsymbol{\tau}, \hat{z})) = -\log \prod_{j=1}^{k^*}\left\{\left(\frac{n_j}{n}\right)^{n_j} p(\{\tau_i\}_{i=1,\hat{z}_i=j}^n; \hat{\mu}_j)\right\} = -n\sum_{j=1}^{k^*}\frac{n_j}{n}\log\frac{n_j}{n} - \sum_{j=1}^{k^*}\log p(\{\tau_i\}_{i=1,\hat{z}_i=j}^n; \hat{\mu}_j)$$

$$= nH\left(\frac{n_1}{n}, \ldots, \frac{n_{k^*}}{n}\right) + \sum_{j=1}^{k^*} n_j \log \hat{\mu}_j + n \tag{2.30}$$

into equation (2.24), where $H(q_1, \ldots, q_{k^*}) = \sum_{j=1}^{k^*} -q_j \log q_j$ represents the entropy, one obtains

$$L_{\mathrm{LVC}}(\boldsymbol{\tau}, \hat{z}) = nH\left(\frac{n_1}{n}, \ldots, \frac{n_{k^*}}{n}\right) + \sum_{j=1}^{k^*} n_j \log \hat{\mu}_j + n + \log C_{\mathrm{EMM}}(n, k^*). \tag{2.31}$$

By substituting equation (2.11) in equation (2.25), one obtains the parametric complexity as follows:

$$C_{\mathrm{EMM}}(n, k^*) = \sum_{\tilde{n}_1 + \cdots + \tilde{n}_{k^*} = n} \frac{n!}{\tilde{n}_1! \cdots \tilde{n}_{k^*}!} \prod_{j=1}^{k^*}\left\{\left(\frac{\tilde{n}_j}{n}\right)^{\tilde{n}_j} C_{\mathrm{EMM}}(\tilde{n}_j, 1)\right\}, \tag{2.32}$$

where

$$C_{\mathrm{EMM}}(n, 1) = \int_{\mathcal{T}(\mu'_{\min}, \mu'_{\max})} p(\boldsymbol{\tau}; \hat{\mu}(\boldsymbol{\tau}))\mathrm{d}\tau_1 \cdots \mathrm{d}\tau_n$$

$$= \left(\frac{n}{e}\right)^n \frac{m'_{\max} - m'_{\min}}{\Gamma(n)}. \tag{2.33}$$

Note that we have used equation (2.21) to derive equation (2.33) and that $C_{\mathrm{EMM}}(n, 1)$ coincides with $C_{\exp}(n)$. Because we cannot calculate $C_{\mathrm{EMM}}(n, k^*)$ using equation (2.32) due to combinatorial explosion, we calculate $C_{\mathrm{EMM}}(n, k^*)$ using the recursive relationship [19] given by

$$C_{\mathrm{EMM}}(n, k+1) = \sum_{r_1 + r_2 = n} \binom{n}{r_1}\left(\frac{r_1}{n}\right)^{r_1}\left(\frac{r_2}{n}\right)^{r_2} C_{\mathrm{EMM}}(r_1, k) C_{\mathrm{EMM}}(r_2, 1). \tag{2.34}$$

The calculation of $C_{\mathrm{EMM}}(n, k^*)$ using this recursive relationship, which dominates the computation time for calculating $L_{\mathrm{LVC}}(\boldsymbol{\tau}, \hat{z})$, requires $O(n^2 k^*)$ time.

Finally, we add the codelength to encode integers $m'_{\min}$ and $m'_{\max}$ to obtain

$$\tilde{L}_{\mathrm{LVC}}(\boldsymbol{\tau}, \hat{z}) = L_{\mathrm{LVC}}(\boldsymbol{\tau}, \hat{z}) + \ell(m'_{\min}) + \ell(m'_{\max}). \tag{2.35}$$

### 2.2.5. Decomposed NML codelength

The second type of the NML codelength with latent variable completion, which we call the decomposed NML (DNML) codelength [20,32], also completes the latent variable and then calculates the NML codelength. The DNML does so after decomposing the joint distribution into the distribution of the latent variables, $z$, and that of the observables, $\boldsymbol{\tau}$, conditioned on the value of $z$ [20]. The DNML codelength was originally formulated for the case in which both observables and latent variables were

discrete [20]. Here, we regard the inter-event time, $\tau_i$, as continuous-valued and the latent variable, $\hat{z}_i$, as discrete-valued.

We start by considering

$$L_{\mathrm{DNML}}(\boldsymbol{\tau},\,\hat{\boldsymbol{z}}) \equiv L_{\mathrm{NML}}(\boldsymbol{\tau}|\hat{\boldsymbol{z}}) + L_{\mathrm{NML}}(\hat{\boldsymbol{z}}). \tag{2.36}$$

Equation (2.22) yields

$$L_{\mathrm{NML}}(\boldsymbol{\tau}|\hat{\boldsymbol{z}}) = \sum_{j=1}^{k^*} L_{\mathrm{NML}}(\{\tau_i\}_{i=1,\hat{z}_i=j}^{n}) = \sum_{j=1}^{k^*} \{n_j \log \hat{\mu}_j + n_j \log n_j - \log \Gamma(n_j)\} + k^* \log(m'_{\max} - m'_{\min}), \tag{2.37}$$

where $m'_{\min}$ and $m'_{\max}$ are given by equations (2.27) and (2.29), respectively. It should be noted that we could prepare integers $m_{\min}$ and $m_{\max}$ for each $j$ ($1 \le j \le k^*$) to code $\{\tau_i\}_{i=1,\hat{z}_i=j}^{n}$ through $\hat{\mu}_j$, similarly to equation (2.23), to derive an alternative of equation (2.37). However, that coding method requires $2k^*$ integers and therefore is less efficient than using only two integers $m'_{\min}$ and $m'_{\max}$.

The NML codelength $L_{\mathrm{NML}}(\hat{\boldsymbol{z}})$ on the right-hand side of equation (2.36) is for a multinomial distribution of $\hat{z}_i$ and is given by

$$L_{\mathrm{NML}}(\hat{\boldsymbol{z}}) = nH\left(\frac{n_1}{n},\, \ldots,\, \frac{n_{k^*}}{n}\right) + \log C_{\mathrm{mult}}(n, k^*). \tag{2.38}$$

In equation (2.38), $C_{\mathrm{mult}}(n, k^*)$ is the parametric complexity for the multinomial distribution having $k^*$ elements. Using equation (2.5), one obtains

$$C_{\mathrm{mult}}(n, k^*) = \sum_{\hat{z}'_1=1}^{k^*} \cdots \sum_{\hat{z}'_n=1}^{k^*} \prod_{j=1}^{k^*} \{\hat{\pi}_j(\hat{z}')\}^{n'_j} = \sum_{\hat{z}'_1=1}^{k^*} \cdots \sum_{\hat{z}'_n=1}^{k^*} \prod_{j=1}^{k^*} \left(\frac{n'_j}{n}\right)^{n'_j}, \tag{2.39}$$

where $n'_j$ is the number of inter-event times whose latent variable $\hat{z}'_i$ is equal to $j$. One can recursively calculate $C_{\mathrm{mult}}(n, k^*)$ [33] by

$$C_{\mathrm{mult}}(n, 1) = 1, \tag{2.40a}$$

$$C_{\mathrm{mult}}(n, 2) = \sum_{t=0}^{n} \frac{n!}{t!(n-t)!} \left(\frac{t}{n}\right)^t \left(\frac{n-t}{n}\right)^{n-t} \tag{2.40b}$$

and

$$C_{\mathrm{mult}}(n, k) = C_{\mathrm{mult}}(n, k-1) + \frac{n}{k-2} C_{\mathrm{mult}}(n, k-2) \quad (k \ge 3). \tag{2.40c}$$

The computational time for solving the set of recursive equations is given by $O(n + k^*)$ and dominates the computational time for $L_{\mathrm{DNML}}(\boldsymbol{\tau}, \hat{\boldsymbol{z}})$.

Finally, similarly to the case of $\mathrm{NML_{LVC}}$, we add the codelength to encode integers $m'_{\min}$ and $m'_{\max}$ to obtain

$$\tilde{L}_{\mathrm{DNML}}(\boldsymbol{\tau}, \hat{\boldsymbol{z}}) = L_{\mathrm{DNML}}(\boldsymbol{\tau}, \hat{\boldsymbol{z}}) + \ell(m'_{\min}) + \ell(m'_{\max}). \tag{2.41}$$

## 2.3. Power-law distributions

We also fitted two types of power-law distributions to the empirical distributions of inter-event times. The first type is the Pareto distribution whose probability density function is given by

$$p(\tau; a, b) = \frac{a-1}{b} \left(\frac{\tau}{b}\right)^{-a}, \tag{2.42}$$

which is defined for $\tau \ge b$. We use the maximum-likelihood estimator given by [34,35]

$$\hat{b}(\boldsymbol{\tau}) = \min_{1 \le i \le n} \tau_i \tag{2.43}$$

and

$$\hat{a}(\boldsymbol{\tau}) = 1 + \frac{1}{(1/n)\sum_{i=1}^{n} \log(\tau_i) - \log(\hat{b})}. \tag{2.44}$$

The second estimator was the one proposed by Clauset *et al.* [36]. In this method, which we refer to as the PLFit algorithm, one selects the $\hat{b}$ value such that the distribution of the data points satisfying $\tau_i \ge \hat{b}$ is as close as possible to a power law. The power-law exponent given the $\hat{b}$ value is

**Table 1.** Properties of the datasets. $N$ represents the number of individuals with at least one inter-event time (therefore, at least two events). The edges are directed in the Bitcoin, Email, College and Sexual datasets. For these datasets, we only considered the individuals as sender but not recipient of the edges. $N_{\geq 100}$ represents the number of individuals with at least 100 inter-event times. The mean and standard deviation (abbreviated as s.d.) are for inter-event times and calculated on the basis of the individuals having at least 100 inter-event times. For the Sexual dataset, $N_{\geq 100}$, the mean and standard deviation are based on the individuals having at least 50 inter-event times.

| data | $N$ | $N_{\geq 100}$ | time resolution | mean $\pm$ s.d. of the inter-event time |
|---|---|---|---|---|
| Office | 92 | 30 | 20 s | $1.17 \times 10^3 \pm 2.56 \times 10^3$ s |
| Hypertext | 112 | 72 | 20 s | $3.91 \times 10^2 \pm 1.11 \times 10^3$ s |
| Reality | 104 | 90 | 1.5 h | $0.50 \pm 1.85$ day |
| Bitcoin | 2969 | 35 | 128 s | $5.05 \times 10^5 \pm 1.58 \times 10^7$ s |
| Email | 777 | 415 | 1 s | $1.26 \times 10^6 \pm 9.49 \times 10^6$ s |
| College | 1176 | 159 | 1 s | $4.18 \times 10^4 \pm 2.55 \times 10^5$ s |
| Sexual | 5660 | 56 | 1 day | $16.9 \pm 24.8$ day |

estimated by equation (2.44). It should be noted that the PLFit does not intend to fit a power-law or different distribution to the data points whose values are less than $\hat{b}$. We used a publicly available Python implementation of the PLFit [37].

# 3. Results

## 3.1. Data

We used the following seven datasets of time-stamped event sequences obtained from human behaviour. Basic properties of each dataset are shown in table 1.

First, we used the data that the SocioPatterns project collected from individuals in an office building that hosted scientific departments [38]. We refer to this dataset as the Office dataset. The individuals wore a sensor on their chest, with which physical proximity between pairs of individuals was detected. The recording lasted for two weeks. A time-stamped event was defined as a contact that lasted at least 20 s, which was the time resolution of the data. For each individual, we ignored the partner of the contact and used the beginning time of each contact as the time of the event. In this manner, we examined a sequence of time-stamped events for each individual in this and the following datasets. Note that, because a contact is a symmetric relationship between two individuals, any event between nodes $i$ and $j$ is considered for both $i$ and $j$.

The inter-event time was defined as the difference between the starting time of the two consecutive events. There was no event during the night-time due to the circadian rhythm. The effect of the circadian rhythm on analysis of inter-event times may be considerable but beyond the scope of the present study. Therefore, we excluded the inter-event times when the two events belonged to different days unless otherwise stated. When there were multiple events in consecutive 20 s time windows between the same pair of individuals, we regarded that they constitute a single event lasting over 20 s rather than a sequence of events with inter-event times equal to 20 s. In practice, in this case, we discarded all but the first event in each sequence of the multiple events in consecutive 20 s time windows and then calculated the inter-event times. We also aggregated events for a focal node that occurred at the same time with different partners into one event. Therefore, the minimum inter-event time is equal to 20 s. For the individuals that have at least 100 inter-event times, the mean and standard deviation of the inter-event times are shown in table 1.

Second, we used the Hypertext 2009 dynamic contact network (Hypertext for short) [39]. Under the SocioPatterns project, the data were collected from approximately 75% of the participants in the HT09 conference in Torino over three days. As was the case for the Office dataset, we ignored inter-event times that span multiple days.

The third dataset originates from the Reality Mining Project (Reality for short), which provides time-stamped physical proximity relationships between the participants of an experiment, who are students or

**Table 2.** Number of components of the EMM selected by each model selection criterion. Each entry of the table shows the selected $(k, k^*)$ pair. For each of the seven datasets, the individual with the largest number of inter-event times is used. The number of inter-event times for that individual is shown in the second column of the table. We compared the EMMs with $k = 1$, 2, …, 9, 10, 20, 50 and 100 under each criterion. For the Sexual dataset, $k = 1$, 2 or 3 were ties when $AIC_{LVC}$, $BIC_{LVC}$, $NML_{LVC}$ or DNML was used. However, these four $k$ values effectively produced the same exponential distribution with $k^* = 1$. Therefore, in the table, we wrote $k = 1$ for these four criteria. Note that, for any criterion, the same $k^*$ value induced by different initial $k$ values may yield different criterion values because the estimated EMM may be different between the different $k$ values even if the $k^*$ value is the same. On the other hand, if different criteria are maximized at the same $k$ value, they are maximized by the same EMM in addition that they share the $k^*$ value.

| data | $n$ | AIC | BIC | $AIC_{LVC}$ | $BIC_{LVC}$ | $NML_{LVC}$ | DNML |
|---|---|---|---|---|---|---|---|
| Office | 403 | (2, 2) | (2, 2) | (2, 2) | (2, 2) | (2, 2) | (2, 2) |
| Hypertext | 659 | (2, 2) | (2, 2) | (2, 2) | (2, 2) | (2, 2) | (2, 2) |
| Reality | 1198 | (3, 3) | (3, 3) | (2, 2) | (2, 2) | (2, 2) | (2, 2) |
| Bitcoin | 604 | (5, 4) | (3, 3) | (4, 3) | (4, 3) | (4, 3) | (4, 3) |
| Email | 3948 | (8, 6) | (8, 6) | (4, 4) | (4, 4) | (4, 4) | (4, 4) |
| College | 1090 | (8, 6) | (4, 4) | (3, 3) | (3, 3) | (3, 3) | (3, 3) |
| Sexual | 118 | (1, 1) | (1, 1) | (1, 1) | (1, 1) | (1, 1) | (1, 1) |

faculty members of MIT. The data were obtained from Bluetooth logs of mobile phones [40]. We did not skip inter-event times overnight because the data were collected over months and the mean inter-event time was much longer than that for the Office and Hypertext datasets (table 1). We used the data from September 2004 to May 2005, which were the months in each of which there were at least $10^3$ events in total.

Fourth, Bitcoin OTC is an over-the-counter marketplace where users trade with bitcoin [41]. In Bitcoin OTC, users rate other users regarding trustworthiness. We neglected the rate score and who a focal user $i$ rated. For each user $i$, we examined a series of time-stamped events, where an event was defined by rating behaviour by user $i$ toward anybody. We refer to this dataset as Bitcoin.

The Email-Eu-core (Email for short) dataset was collected from a large European research institution between October 2003 and May 2005 [42]. A node was an email address. An edge was defined between two nodes if and only if they sent email to each other at least once. A time-stamped event for node $i$ in the network was defined by an email that $i$ sent to anybody, disregarding the identity of the recipient of the email.

The CollegeMsg (College for short) data were collected from students belonging to an online community at the University of California, Irvine, between April and December 2004 [43]. The event was defined as the message sent from a user to another user. We downloaded the Bitcoin, Email and College data from the Stanford Network Analysis Project (SNAP) website (http://snap.stanford.edu/).

The sexual contact (Sexual for short) dataset provides times of online sexual encounters between escorts and sex buyers [44]. We used event sequences for each sex buyer by regarding a commercial sexual activity with any escort as an event.

## 3.2. Fitting of the EMM to the individual with the largest number of events and comparison with power-law distributions

We fitted EMMs with different numbers of components, $k$, to the sequence of inter-event times obtained from each individual in each dataset. We examined $k = 1, 2, …, 9, 10, 20, 50$ and 100 in each case. Then, according to each of the six model selection criteria, AIC, BIC, $AIC_{LVC}$, $BIC_{LVC}$, $NML_{LVC}$ and DNML, we selected the best $k$ value.

Results of model selection for the individual with the largest number of events in each dataset are shown in table 2. Table 2 indicates that the effective number of components finally chosen, $k^*$, is at most four in all but three out of the 42 combinations of the dataset and criterion. Furthermore, for four out of the seven datasets, $k^*$ is at most three regardless of the criterion. These results suggest that a mixture of a small number of exponential distributions may be a reasonable approximation to empirical distributions of inter-event times in many cases.

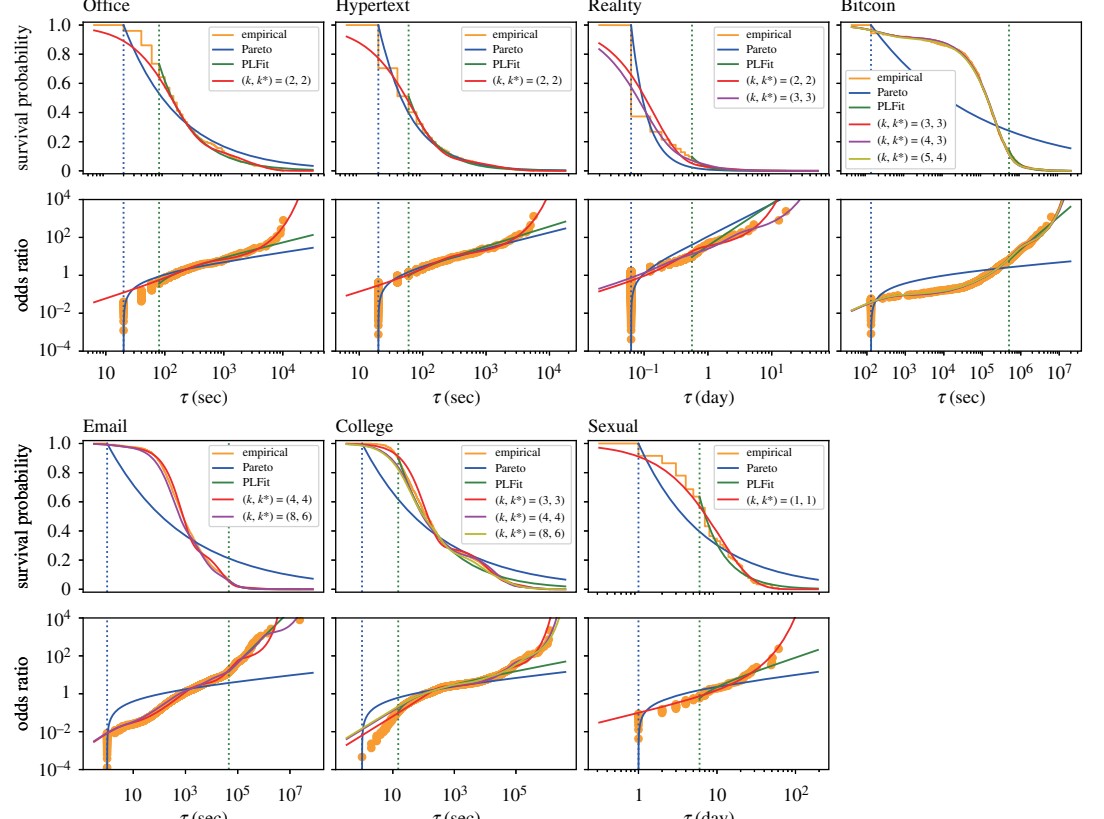

**Figure 1.** Survival probability and odds ratio of inter-event times for the individual with the largest number of events in each dataset. The empirical and estimated distributions are compared. For the EMMs, the distributions for the $(k, k^*)$ pairs selected under at least one model selection criterion are shown. For the Bitcoin data, the three selected EMMs with different $k$ and $k^*$ values almost overlap each other. The estimated parameter values for the selected EMMs are shown in the electronic supplementary material. The dotted vertical lines represent the estimated lower bounds, $\hat{b}$, for the two types of power-law distributions. Because the PLFit fits a power law to the data points with $\tau_i \geq \hat{b}$, we rescaled the estimated survival probabilities by a factor of $n'/n$, where $n'$ is the number of inter-event times satisfying $\tau_i \geq \hat{b}$. Similarly, we normalized the odds ratio for the same power-law distribution by $OR(\tau) = [1 - (n'/n) \int_\tau^\infty p(\tau') d\tau'] / [(n'/n) \int_\tau^\infty p(\tau') d\tau']$.

Next, we compared the performance between the estimated EMMs and power-law estimators in approximating the empirical distributions. We have fitted power-law distributions because they have widely been applied to empirical distributions of inter-event times [1–4]. For the same individuals as those used in table 2, the empirical and estimated distributions of inter-event times are compared in figure 1. When the different model selection criteria yielded different EMMs (i.e. EMMs with different values of $k$ or $k^*$), we showed all the selected EMMs overlaid in the figure. We compared the different distributions in terms of the survival probability (i.e. the probability with which the inter-event time is larger than $\tau$, i.e. $\int_\tau^\infty p(\tau') d\tau'$) and the odds ratio defined by $OR(\tau) = [1 - \int_\tau^\infty p(\tau') d\tau'] / \int_\tau^\infty p(\tau') d\tau'$. Note that the odds ratio is particularly good at capturing differences between distributions at small values of $\tau$ [45].

Figure 1 elicits the following observations. First, in five datasets (i.e. except Bitcoin and Email) the right tails of the distribution seem to be more accurately approximated by an EMM than the power-law distributions. This may be because the two power-law estimators employed here assume a strictly power-law tail, whereas empirical data usually have exponential cut-offs. For the Bitcoin and Email datasets, the power-law estimate obtained by the PLFit algorithm seems to be roughly as accurate as EMMs in approximating right tails of the distribution. Second, the odds ratio plots show that, at small $\tau$ values, the approximation looks the most accurate with the Pareto distribution in all but the College dataset.

To examine whether these casual observations extend to the entire set of inter-event times from the same individuals, we investigated the likelihood of the empirical data in each model. The likelihood for

**Table 3.** Likelihood of the entire and truncated datasets. The results for the individual with the largest number of inter-event times in each dataset are shown. The number of inter-event times larger than $\min_{i=1}^{n} \tau_i$ and $\hat{b}$ as estimated by the PLFit is denoted by $n''$ and $n'$, respectively. The columns labelled all, $\tau > \min_i \tau_i$, and $\tau \geq \hat{b}$ are the likelihood values for the $n$, $n''$ and $n'$ inter-event times, respectively.

| data | all | | | $\tau > \min \tau_i$ | | |
|---|---|---|---|---|---|---|
| | $n$ | EMM | Pareto | $n''$ | EMM | Pareto |
| Office | 403 | −2780.8 | −2804.4 | 387(96%) | −2696.3 | −2744.0 |
| Hypertext | 659 | −3826.3 | −3534.8 | 464(70%) | −2924.6 | −2916.4 |
| Reality | 1198 | 843.4 | 2063.4 | 447(37%) | −341.0 | −420.8 |
| Bitcoin | 604 | −7968.0 | −8519.7 | 570(94%) | −7680.8 | −8291.7 |
| Email | 3948 | −36 058.3 | −38 853.4 | 3915(99%) | −35 893.1 | −38 789.6 |
| College | 1090 | −8621.8 | −9073.1 | 1089($\approx$100%) | −8617.5 | −9071.4 |
| Sexual | 118 | −396.9 | −423.4 | 108(92%) | −372.3 | −416.8 |

| data | $\tau \geq \hat{b}$ | | | |
|---|---|---|---|---|
| | $n'$ | EMM | Pareto | PLFit |
| Office | 296(73%) | −2196.1 | −2278.2 | −2056.7 |
| Hypertext | 337(51%) | −2301.1 | −2351.7 | −2034.6 |
| Reality | 116(10%) | −346.6 | −448.3 | −9.4 |
| Bitcoin | 85(14%) | −1387.0 | −1438.8 | −1202.0 |
| Email | 253(6%) | −3799.7 | −3809.0 | −3075.2 |
| College | 983(90%) | −8140.2 | −8616.6 | −8021.2 |
| Sexual | 75(64%) | −283.4 | −334.3 | −237.2 |

the three models, i.e. the selected EMM, maximum-likelihood estimator of the Pareto distribution, and PLFit, is compared in table 3. Because the PLFit discards small inter-event times, we start by comparing the EMM and Pareto distribution when all inter-event times are used. The table indicates that the EMM realizes a larger likelihood value than the Pareto distribution in five datasets and vice versa in the other two datasets (i.e. Hypertext and Reality). The EMM has more parameters than the Pareto distribution, which one may suspect as a reason why the EMM fits the data better than the Pareto distribution in more datasets than vice versa. However, the results are qualitatively the same when the two models are compared in terms of the AIC and BIC (electronic supplementary material, table S1). We remark that the maximum-likelihood estimator for the Pareto distribution is not asymptotically normal such that the application of the AIC and BIC to the Pareto distribution as well as to EMMs is not justified. It should be noted that the other four criteria are not relevant to the Pareto distribution because it is not a latent variable model.

For the Hypertext and Reality datasets, for which the likelihood is larger for the Pareto distribution than the optimal EMM, there are relatively many data points with $\tau_i = \tau_{\min}$ (table 3). The Pareto distribution beats the EMM probably because it avoids devoting the probability mass to values less than $\tau_{\min}$ by definition, and it is accurate at many data points that have $\tau_i = \tau_{\min}$. To clarify this point, we compared the likelihood when the data points with $\tau_i = \tau_{\min}$ were excluded. The results are shown in table 3. To simplify the discussion, for the EMM we employed the value of $k$ that minimized the AIC. For the Reality data, the EMM fits the data better than the Pareto distribution when we only consider the inter-event times larger than $\tau_{\min}$. For the Hypertext dataset, the Pareto distribution is still better than the EMM, but the difference in the likelihood is much smaller than when all the inter-event times are used. Note that the Reality dataset has a larger fraction of inter-event times with $\tau = \tau_{\min}$ than the Hypertext dataset (table 3).

The PLFit only fits a power-law distribution to relatively large inter-event times. Therefore, we compared the likelihood of the EMM, Pareto distribution, and PLFit for the set of inter-event time values satisfying $\tau_i \geq \hat{b}$, where $\hat{b}$ is the threshold estimated by the PLFit. The PLFit is the best

performer among the three fitting schemes (table 3). This is probably because the threshold is optimized for the PLFit in this comparison and the PLFit is allowed to discard small inter-event times. By contrast, the EMM intends to fit the entire set of inter-event times. Furthermore, in this comparison, the EMM is disadvantageous by the amount of probability mass that it has to devote to inter-event times smaller than $\hat{b}$. It should be noted that the PLFit discards a considerable fraction of data points, ranging between 27 and 94%, except for the College dataset (table 3). It should also be noted that the EMM yields a larger likelihood value than the Pareto distribution for all the datasets with this $\hat{b}$ value.

We do not conclude whether or not the EMM outperforms the Pareto distribution or PLFit. Both the Pareto distribution and PLFit benefit from carefully setting the lower bound on the inter-event times, $\hat{b}$. When all inter-event time data have to be modelled and we do not know the smallest possible value of the inter-event time, EMMs may be preferred to the Pareto distribution and the PLFit.

So far, we have excluded the inter-event times across consecutive days in the Office and Hypertext datasets. We also ran the same analysis for these two datasets without excluding such long inter-event times. The results were qualitatively the same except that the EMM estimated the number of components larger by one (electronic supplementary material). The new component (i.e. a single exponential distribution in the EMM) corresponded to the across-day inter-event times, with a considerably longer mean inter-event time than that for the other estimated components. Furthermore, the mean inter-event time of the constituent exponential distributions that exist both when across-day inter-event times are included and when excluded are almost the same in the two cases (electronic supplementary material). We conclude that inter-event times on the timescale of a day or longer contribute one exponential distribution with a large mean to the entire EMM without interfering with the constituent exponential distributions on smaller time scales.

## 3.3. Population results

In the previous section, we showed that mixtures of a small number of exponential distributions tended to be selected according to the different model selection criteria. To assess the generality of this result, we carried out the model selection for all individuals having at least 100 inter-event times. Because the Sexual dataset has only two individuals with at least 100 inter-event times, we instead used the 56 individuals that had at least 50 inter-event times for this dataset. The histogram of the optimal $k^*$ value for each combination of the dataset and criterion is shown in figure 2. Consistently with the results for the individuals with the largest number of inter-event times in each dataset, the effective number of components, $k^*$, was estimated to be at most three in four out of the seven datasets (i.e. Office, Hypertext, Reality and Sexual). For the Bitcoin dataset, the distributions of $k^*$ had the mode at 3, and we obtained $2 \leq k^* \leq 4$ in most cases. For the other two datasets (i.e. Email and College), the AIC and BIC yielded distributions of $k^*$ whose mode was four and the largest value was seven. Under the other four criteria, which are the ones justified for EMMs, the mode was equal to three, and the largest $k^*$ was equal to five.

To exclude the possibility that longer data (i.e. larger $n$) tend to yield a larger $k^*$ value, we calculated the Pearson correlation coefficient between $n$ and the optimal $k^*$ for each combination of the dataset and criterion. To calculate the Pearson correlation coefficient, we only used the individuals with at least 100 inter-event times (and at least 50 inter-event times for the Sexual dataset), similar to figure 2. The Pearson correlation coefficient values, denoted by $r$, and its $p$-value are shown in table 4. The correlation coefficient was small and insignificant except for the Email dataset combined with all criteria and the College dataset combined with the AIC or BIC. When we confined the analysis to the individuals with larger numbers of inter-event times, i.e. at least 500 and 200 inter-event times for the Email and College datasets, respectively, the Pearson correlation coefficient was smaller. In particular, for the $AIC_{LVC}$, $BIC_{LVC}$, $NML_{LVC}$ and DNML, the correlation was close to zero and insignificant when we only considered the individuals with many inter-event times. Note that the largest number of inter-event times for these two datasets were sufficiently larger than the new threshold values (Email: $n = 3948$, College: $n = 1090$) and that there remained sufficiently many individuals with the new threshold values (Email: 117 individuals, College: 61 individuals). The results suggest that our main result that the optimal number of components, $k^*$, tends to be small is expected to remain true for long sequences of inter-event times.

## 4. Discussion

We showed that EMMs (i.e. mixtures of exponential distributions) with a small number of components, up to three in many cases and four in some cases, were a reasonably good fit to various empirical

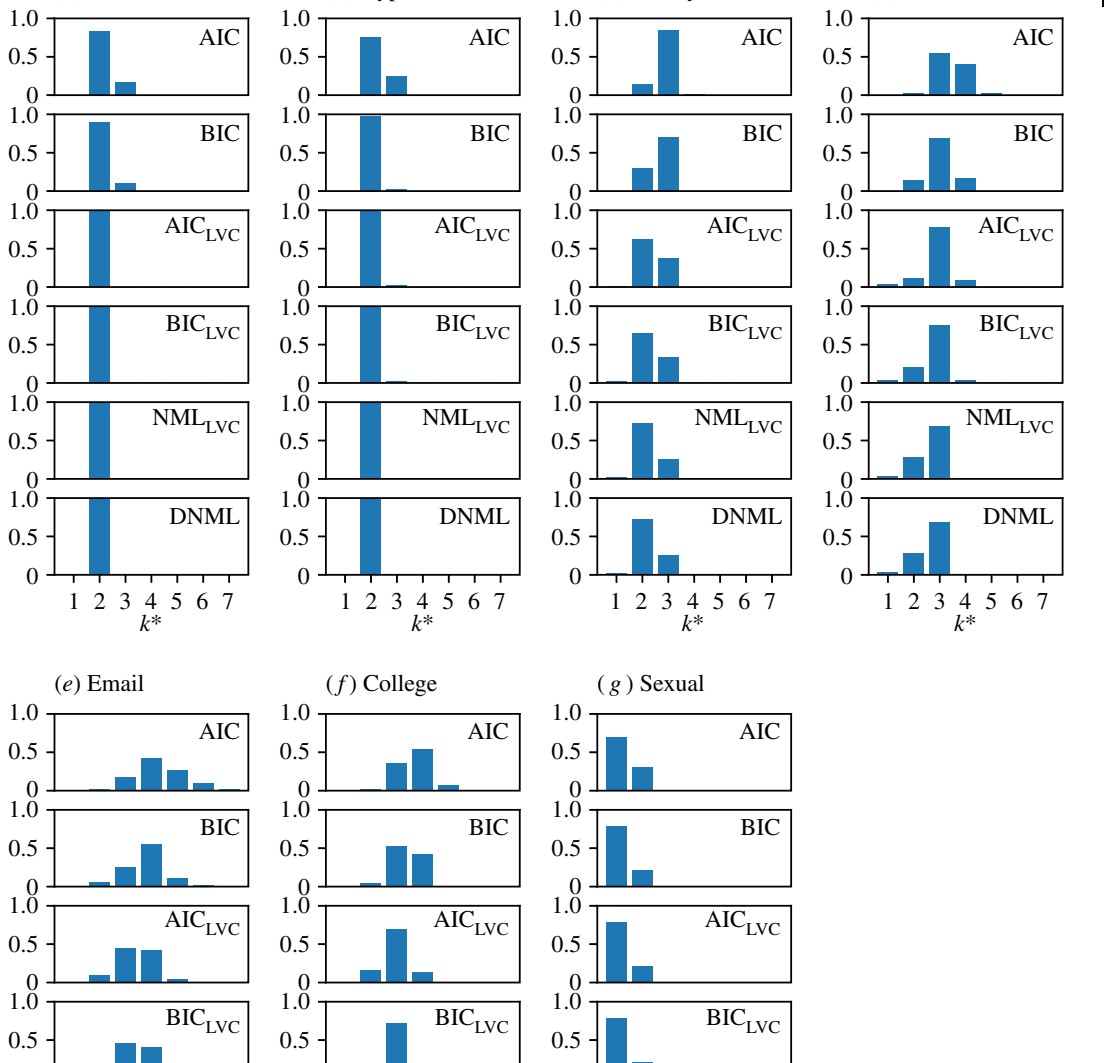

**Figure 2.** Distributions of the effective number of components in the EMM, $k^*$. (a) Office, (b) Hypertext, (c) Reality, (d) Bitcoin, (e) Email, (f) College and (g) Sexual. For the AIC and BIC, the model selection is carried out in terms of $k$, and the $k^*$ values shown in this figure are those corresponding to the selected $k$ values. We calculated the distributions on the basis of the individuals with at least 100 inter-event times with the exception of the Sexual dataset, for which we used the individuals with at least 50 inter-event times (see table 1 for the number of such individuals).

distributions of inter-event times. In general, there are various mechanisms behind power-law distributions of an observable, and many of them are believed to be relevant as generative mechanisms of long-tailed degree distributions of empirical networks [46]. By contrast, the present results suggest that empirical long-tailed distributions can be also approximated by EMMs. Furthermore, EMMs can be obtained as an outcome of stochastic point processes in which the system's state switches between a small number of discrete states in each of which the process generates events as a Poisson process. In practice, a human or animal individual may maintain a

**Table 4.** Relationship between the best $k^*$ value and the number of inter-event times (i.e. $n$) across the individuals within each dataset. We calculated the Pearson correlation coefficient, $r$, between $k^*$ and $n$, and its $p$-value. The calculation was based on the individuals having at least 100 inter-event times unless we state the threshold value in the first column. The cells without results are the cases in which all the individuals yielded the same $k^*$ value and hence one cannot calculate the correlation coefficient.

| data | | AIC | BIC | $AIC_{LVC}$ | $BIC_{LVC}$ | $NML_{LVC}$ | DNML |
|---|---|---|---|---|---|---|---|
| Office | $r$ | $-0.077$ | $-0.078$ | — | — | — | — |
| | $p$ | $6.9 \times 10^{-1}$ | $6.8 \times 10^{-1}$ | — | — | — | — |
| Hypertext | $r$ | $0.005$ | $-0.034$ | $-0.004$ | $-0.004$ | — | — |
| | $p$ | $9.6 \times 10^{-1}$ | $7.8 \times 10^{-1}$ | $9.7 \times 10^{-1}$ | $9.7 \times 10^{-1}$ | — | — |
| Reality | $r$ | $-0.049$ | $-0.097$ | $-0.015$ | $-0.018$ | $-0.036$ | $-0.036$ |
| | $p$ | $6.5 \times 10^{-1}$ | $3.6 \times 10^{-1}$ | $8.9 \times 10^{-1}$ | $8.7 \times 10^{-1}$ | $7.4 \times 10^{-1}$ | $7.4 \times 10^{-1}$ |
| Bitcoin | $r$ | $0.230$ | $0.011$ | $0.041$ | $0.006$ | $0.152$ | $0.152$ |
| | $p$ | $1.8 \times 10^{-1}$ | $9.5 \times 10^{-1}$ | $8.2 \times 10^{-1}$ | $9.7 \times 10^{-1}$ | $3.8 \times 10^{-1}$ | $3.8 \times 10^{-1}$ |
| Email | $r$ | $0.490$ | $0.470$ | $0.239$ | $0.226$ | $0.229$ | $0.243$ |
| | $p$ | $1.8 \times 10^{-26}$ | $3.5 \times 10^{-24}$ | $8.0 \times 10^{-7}$ | $3.5 \times 10^{-6}$ | $2.4 \times 10^{-6}$ | $5.2 \times 10^{-7}$ |
| College | $r$ | $0.392$ | $0.345$ | $0.106$ | $0.101$ | $0.110$ | $0.110$ |
| | $p$ | $3.2 \times 10^{-7}$ | $8.4 \times 10^{-6}$ | $1.8 \times 10^{-1}$ | $2.0 \times 10^{-1}$ | $1.7 \times 10^{-1}$ | $1.7 \times 10^{-1}$ |
| Sexual ($\geq 50$) | $r$ | $0.035$ | $-0.049$ | $-0.012$ | $-0.012$ | $0.162$ | $0.099$ |
| | $p$ | $8.0 \times 10^{-1}$ | $7.2 \times 10^{-1}$ | $9.3 \times 10^{-1}$ | $9.3 \times 10^{-1}$ | $2.3 \times 10^{-1}$ | $4.7 \times 10^{-1}$ |
| Email ($\geq 500$) | $r$ | $0.454$ | $0.331$ | $0.073$ | $0.061$ | $0.022$ | $0.058$ |
| | $p$ | $2.7 \times 10^{-7}$ | $2.7 \times 10^{-4}$ | $4.3 \times 10^{-1}$ | $5.1 \times 10^{-1}$ | $8.1 \times 10^{-1}$ | $5.3 \times 10^{-1}$ |
| College ($\geq 200$) | $r$ | $0.313$ | $0.261$ | $0.004$ | $0.002$ | $0.011$ | $0.011$ |
| | $p$ | $1.4 \times 10^{-2}$ | $4.2 \times 10^{-2}$ | $9.8 \times 10^{-1}$ | $9.9 \times 10^{-1}$ | $9.3 \times 10^{-1}$ | $9.3 \times 10^{-1}$ |

small number of relatively discrete states, such as active and rest. The event rate may depend on the discrete state. It should be noted that, within each state, the mechanism to generate event sequences is maximally simple: Poisson process with a constant rate. To the best of our knowledge, Poissonian views of long-tailed distributions of inter-event times were first introduced in [5]. The contribution of the present work is to have introduced a formal model selection framework to directly compare EMMs with different numbers of components and having carried out some comparisons with power-law distributions.

There is a recent debate regarding the abundance of scale-free (i.e. power-law) degree distributions in empirical networks. When a pure power-law tail is assumed for the degree distribution, there are empirically few scale-free networks [47]. By contrast, scale-free degree distributions are abundant when impurity in the power-law distributions via the regularly varying distributions is considered [48]. In the present study, we fitted the pure power-law distributions via the PLFit algorithm [36]. The PLFit was more accurate than EMMs while the PLFit discarded small inter-event times, which occupied a non-negligible fraction of inter-event times in each dataset. We did not examine the method presented in [48]. This is because the purpose of the present study was not to find an accurate estimator or to test if a power-law distribution of inter-event times was a good fit. Rather, our purpose was to test the hypothesis that the system generating time-stamped events, such as humans, transit between a small number of discrete states each of which is a Poisson process generator.

A reasonably accurate fit of the EMMs to the empirical data revealed in the present study mainly comes from a high accuracy at small $\tau$ values, because there are many data points with small $\tau$. Such short inter-event times yield a component exponential distribution with a small mean inter-event time and may be produced as a result of the visit of the individual to an active state. Given these considerations, looking at the entire distribution of the data rather than focusing on the distribution's tail may help us to understand mechanisms generating the data.

The authors of [10] reached a similar conclusion to ours, where they split the inter-event times into two groups by thresholding. Then, they fitted an exponential distribution to the inter-event times exceeding the threshold and another exponential distribution to the inter-event times less than the

threshold. Their models are EMMs with two components. In contrast to their study, we systematically fitted EMMs and inferred the number of components using model selection criteria valid for EMMs. Testing our method on various other distributions of inter-event times to clarify the reason why EMMs with a small number of components apply accurately to some datasets and not others warrants future work.

Data accessibility. The data are open resources. Python codes for estimating and selecting the mixture of exponential distributions are available at Github (https://github.com/naokimas/exp_mixture_model).

Authors' contributions. K.Y. and N.M. conceived the study. N.M. designed the study. M.O. carried out the theoretical analysis and performed the data analysis. M.O., K.Y. and N.M. discussed the results and drafted the manuscript. All authors read and approved the manuscript.

Competing interests. The authors declare that they have no competing interests.

Funding. K.Y. and N.M. acknowledge the support provided through JST CREST grant no. JPMJCR1304, Japan.

Acknowledgements. We thank the SocioPatterns collaboration (http://www.sociopatterns.org) for providing the two datasets used in the present paper.

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
