## [Reviewer comments · Royal Society Open Science]

Review History

RSOS-191643.R0 (Original submission)

Review form: Reviewer 1

Is the manuscript scientifically sound in its present form?

No

Are the interpretations and conclusions justified by the results?

Yes

Is the language acceptable?

Yes

Do you have any ethical concerns with this paper?

No

Have you any concerns about statistical analyses in this paper?

No

Recommendation?

Major revision is needed (please make suggestions in comments)

Comments to the Author(s)

The title of the article is 'Long-tailed distributions of inter-event times as mixtures of exponential distributions' and it first reading it suggests that log-tailed distribtuon is approximated with exponential mixture. Contrary to this, the article major focus is on the model selection assuming mixture of exponential distributions. The problem is well studied in the literature and the present article lacks the novelty or contribution is marginal. The authors must fit some other generalized distributions to the data asuming renewal process which is a more general stochastic process than the Poisson process.

How the proposed methodology is applicable to shifted exponential distribution?

How did you cope the problem of label switching?

From a computational perspective, the data are usually sparse, so it is amenable for algorithms that are cost-effective. How the present approach is cost effective in such cases.?

Review form: Reviewer 2

Is the manuscript scientifically sound in its present form?

Yes

Are the interpretations and conclusions justified by the results?

Yes

Is the language acceptable?

Yes

Do you have any ethical concerns with this paper?

No

Have you any concerns about statistical analyses in this paper?

No

Recommendation?

Accept as is

Comments to the Author(s)

The paper is well-arranged and clearly written.
Recommended for publication.

Decision letter (RSOS-191643.R0)

03-Dec-2019

Dear Dr Masuda,

The editors assigned to your paper ("Long-tailed distributions of inter-event times as mixtures of exponential distributions") have now received comments from reviewers. We would like you to

revise your paper in accordance with the referee and Associate Editor suggestions which can be found below (not including confidential reports to the Editor). Please note this decision does not guarantee eventual acceptance.

Please submit a copy of your revised paper before 26-Dec-2019. Please note that the revision deadline will expire at 00.00am on this date. If we do not hear from you within this time then it will be assumed that the paper has been withdrawn. In exceptional circumstances, extensions may be possible if agreed with the Editorial Office in advance. We do not allow multiple rounds of revision so we urge you to make every effort to fully address all of the comments at this stage. If deemed necessary by the Editors, your manuscript will be sent back to one or more of the original reviewers for assessment. If the original reviewers are not available, we may invite new reviewers.

- Data accessibility

<http://datadryad.org/submit?journalID=RSOS&manu=RSOS-191643>

- Competing interests

- Authors' contributions

All submissions, other than those with a single author, must include an Authors' Contributions section which individually lists the specific contribution of each author. The list of Authors

should meet all of the following criteria; 1) substantial contributions to conception and design, or acquisition of data, or analysis and interpretation of data; 2) drafting the article or revising it critically for important intellectual content; and 3) final approval of the version to be published.

- Acknowledgements

- Funding statement

Kind regards,

on behalf of Professor Weisi Guo (Associate Editor) and Mark Chaplain (Subject Editor)
 openscience@royalsociety.org

Reviewers' Comments to Author:

Reviewer: 1

Comments to the Author(s)

The title of the article is 'Long-tailed distributions of inter-event times as mixtures of exponential distributions' and it first reading it suggests that log-tailed distribution is approximated with exponential mixture. Contrary to this, the article major focus is on the model selection assuming mixture of exponential distributions. The problem is well studied in the literature and the present article lacks the novelty or contribution is marginal. The authors must fit some other generalized distributions to the data assuming renewal process which is a more general stochastic process than the Poisson process.

How the proposed methodology is applicable to shifted exponential distribution?

How did you cope the problem of label switching?

From a computational perspective, the data are usually sparse, so it is amenable for algorithms that are cost-effective. How the present approach is cost effective in such cases.?

Reviewer: 2
Comments to the Author(s)

The paper is well-arranged and clearly written.
Recommended for publication.

Author's Response to Decision Letter for (RSOS-191643.R0)

See Appendix A.

RSOS-191643.R1 (Revision)

Review form: Reviewer 1

Is the manuscript scientifically sound in its present form?
Yes

Are the interpretations and conclusions justified by the results?
Yes

Is the language acceptable?
Yes

Do you have any ethical concerns with this paper?
No

Have you any concerns about statistical analyses in this paper?
No

Recommendation?
Accept as is

Comments to the Author(s)
OK

Decision letter (RSOS-191643.R1)

28-Jan-2020

Dear Dr Masuda,

It is a pleasure to accept your manuscript entitled "Long-tailed distributions of inter-event times as mixtures of exponential distributions" in its current form for publication in Royal Society Open Science. The comments of the reviewer(s) who reviewed your manuscript are included at the foot of this letter.

At this stage, we ask that you please archive your GitHub code within the Zenodo repository: <https://guides.github.com/activities/citable-code/>. By doing this, a formal, citable DOI will be associated with your data record, and an open license (CC-BY preferred) can be applied to your data. We would then update your data availability statement to read as:

"Data and relevant code for this research work are stored in GitHub: [GitHub URL here] and have been archived within the Zenodo repository: <https://doi.org/zenodo.....> [ref number]"

Please confirm the details of the accession as soon as possible.

on behalf of Professor Weisi Guo (Associate Editor) and Mark Chaplain (Subject Editor)
openscience@royalsociety.org

Reviewer comments to Author:
Reviewer: 1

Comments to the Author(s)
OK

Appendix A

Response to Reviewer 1

The title of the article is ‘Long-tailed distributions of inter-event times as mixtures of exponential distributions’ and it first reading it suggests that log-tailed distribution is approximated with exponential mixture. Contrary to this, the article major focus is on the model selection assuming mixture of exponential distributions. The problem is well studied in the literature and the present article lacks the novelty or contribution is marginal. The authors must fit some other generalized distributions to the data assuming renewal process which is a more general stochastic process than the Poisson process.

In the previous version, we had already fitted two power-law distributions assuming renewal processes and compared with the fitting by mixtures of exponential distributions. Most of section 3(b) (i.e., from the third paragraph to the second last paragraph) is devoted to it.

How the proposed methodology is applicable to shifted exponential distribution?

It is possible to consider mixtures of shifted exponential distributions. However, we do not see a particular reason why we should consider this distribution. First, a mixture of shifted exponential distributions has approximately twice more parameters than an EMM (i.e., mixture of exponential distributions without shift). Therefore, the model estimation will be more involved than that for EMMs. Second, we do not know a rationale behind assuming (a mixture of) shifted exponential distributions as a model of human behavior. An exponential distribution (i.e., Poisson process) models memoryless event generation processes. An EMM models the situation in which a human individual switches between, for example, an active and inactive state in the case of $k = 2$ components, and produces events in a memoryless manner in each of the states, as motivated in the main text. We also fitted power law distributions because many previous researches fitted them to data, and so we should also use them as benchmark. We do not see a similar rationale for shifted exponential distributions or other distributions, while we are not claiming that they should not fit well to data. We also consider that discussing shifted exponential distributions in the main text (in the Discussion section)

will also confuse readers for the same reason. Therefore, we opt not to change the manuscript in this respect.

How did you cope the problem of label switching?

We obtained the hidden labels, z_i , through maximization of the posterior distribution induced by the EM algorithm (rather than sampling with Monte Carlo Markov chain). This procedure does not involve any sampling. In the present manuscript, we did not use the obtained maximum likelihood estimator, including the estimated labels, in a way that requires sampling, either. The label switching is a problem inherent in sampling; point estimators such as the maximum likelihood estimator do not suffer from the problem (e.g., Puolamäki & Kaski, LNCS, 5772, 381–392, 2009, doi:10.1007/978-3-642-03915-7_33).

From a computational perspective, the data are usually sparse, so it is amenable for algorithms that are cost-effective. How the present approach is cost effective in such cases.?

We are estimating distribution of inter-event times, no matter whether it is in fact close to a power-law distribution or not. Therefore, there is no sparse data involved.

Finally, we corrected a typo in Abstract, i.e., removed “distributions of” in line 7 in Abstract.